# Catecholaminergic modulation of the cost of cognitive control in healthy older adults

**Monja I. Froböse**[1,2]*, **Andrew Westbrook**[1,3], **Mirjam Bloemendaal**[1], **Esther Aarts**[1], **Roshan Cools**[1,4]

**1** Donders Institute for Brain, Cognition and Behaviour, Radboud University, Nijmegen, The Netherlands, **2** Institute of Experimental Psychology, Heinrich-Heine University, Düsseldorf, Germany, **3** Cognitive, Linguistic, and Psychological Sciences, Brown University, Providence, RI, United States of America, **4** Dept Psychiatry, Radboud University Medical Centre, Nijmegen, The Netherlands

* m.i.froboese@gmail.com

**Data Availability Statement:** Data and code are freely available via the Open Science Framework (https://osf.io/n5xev/).

**Funding:** Roshan Cools (RC) James McDonnell Scholar Award (220020328) from James C. McDonnell Foundation (https://www.jsmf.org/).

## Abstract

Catecholamines have long been associated with cognitive control and value-based decision-making. More recently, we have shown that catecholamines also modulate value-based decision-making about whether or not to engage in cognitive control. Yet it is unclear whether catecholamines influence these decisions by altering the subjective value of control. Thus, we tested whether tyrosine, a catecholamine precursor altered the subjective value of performing a demanding working memory task among healthy older adults (60–75 years). Contrary to our prediction, tyrosine administration did not significantly increase the subjective value of conducting an N-back task for reward, as a main effect. Instead, in line with our previous study, exploratory analyses indicated that drug effects varied as a function of participants' trait impulsivity scores. Specifically, tyrosine increased the subjective value of conducting an N-back task in low impulsive participants, while reducing its value in more impulsive participants. One implication of these findings is that the over-the-counter tyrosine supplements may be accompanied by an undermining effect on the motivation to perform demanding cognitive tasks, at least in certain older adults. Taken together, these findings indicate that catecholamines can alter cognitive control by modulating motivation (rather than just the ability) to exert cognitive control.

## 1 Introduction

While catecholamines (dopamine and noradrenaline) have long been known to impact capacity for cognitive control, the catecholamines have been proposed to also mediate cost-benefit choices about whether or not to exert cognitive control [1,2]. According to the expected value of control account, people recruit cognitive control in proportion to expected instrumental value [3], such that degree (and intensity) of engagement in an upcoming cognitive computation is based on a cost-benefit analysis. Recently, we demonstrated that acute administration of a single oral dose of the catecholamine transporter blocker methylphenidate indeed modulated the avoidance of, but not ability to perform cognitive control in young adults [4]. The effect depended on trait impulsivity, with the most impulsive subjects exhibiting the greatest

Vici Award (453.14.015) from Nederlandse Organisatie voor Wetenschappelijk Onderzoek (NWO; https://www.nwo.nl/en). Esther Aarts (EA) European Regional Development Fund and the Dutch provinces Gelderland and Overijssel Grant 2011-017004 (FOCOM; http://www.focom-project.ruhosting.nl/about/). Sponsors did not play roles in study design, data collection and analysis, decision to publish or preparation of manuscript.

**Competing interests:** The authors declare no competing financial interest.

increases in control avoidance. Here, we extend this work by assessing the effects of a catecholamine precursor on the expected value of cognitive control, again as a function of trait impulsivity.

## 1.1 Catecholamines and (cognitive) effort

The role of catecholamines in decisions about effort expenditure have been the focus of studies for decades [5]. A well-replicated finding, in both human and non-human animals is that striatal dopamine blockade or dopamine lesions reliably shift preferences away from high effort/high reward options to low effort/low reward options [e.g. 5,6], while increases in striatal dopamine shifts preferences towards high effort/high reward options [5,7–9]. For example, patients with Parkinson's disease, which is characterized by dopamine cell loss in the striatum, forego reward to avoid effort (handgrip squeezes) relative to healthy controls when tested off their dopaminergic medication. Conversely, when tested on their medication, patients selected high-effort/high-reward options as much as controls, reflecting less physical effort avoidance [7]. Thus, increases in dopamine transmission have been associated with increased motivation for physical effort.

As is the case for physical action, cognitive control is also effortful / costly such that people tend to avoid it [2,3,10,11]. For example, they prefer to perform a task with less rather than more task-switching [12–15] and with lower rather than higher working memory load, even when incentives are larger for higher loads [16]. However, unlike for physical effort, the role of catecholamines is less clear for decision making about cognitive effort and it is still under study whether findings regarding physical effort valuation (entirely) generalize to the cognitive domain [17–19]. One way in which dopamine might bias choices about cognitive tasks is by altering the (expected) value of cognitive control (i.e. the reward benefits minus the effort cost of control). This follows also from neurocomputational models of dopamine in the basal ganglia, such as the OpAL model, which suggest that increases in (striatal) dopamine tone, result in more emphasis on the benefit, and less on the cost of an action due to more direct pathway excitability, via D1 receptor binding, and less indirect pathway excitability, via D2 receptor binding [20]. Thus, taken together with the hypothesis that cognitive control follows from cost-benefit decision making, we expect that increases in catecholamine synthesis will emphasize the benefits versus the costs of control, thereby increasing the motivation for instrumental cognitive control [2].

There is some evidence that dopamine signaling can offset the costs of cognitive control. In one study, costs were offset by incentives, which are putatively signaled by dopamine release, thus leading to more cognitive and motor control in a visual saccade task [21]. Critically, this effect was diminished in patients with Parkinson's disease, supporting a role for dopamine in mediating incentives effects on performance. Since performance could be altered via multiple catecholamine-dependent mechanisms, however, it remains critical to show that catecholaminergic drugs can alter cost-benefit decision-making itself. Direct tests of this prediction have yielded conflicting results. In one study, dopaminergic medication increased the selection of high-cognitive effort/high-benefit tasks in Parkinson's disease [22]. By contrast, a rodent study failed to observe changes in rats' willingness to expend cognitive effort for reward after treatment with a dopamine antagonist [17].

Conflicting results may stem from individual differences in baseline dopamine function and/or cognitive motivation. In one study, amphetamine motivated rodent 'slackers' (but not 'workers') to choose a more perceptually-demanding option for a higher reward [23]. In parallel, our recent work with young healthy adults has shown that the administration of methylphenidate (20 mg, oral) altered the avoidance of higher task-switching demands, in a demand

selection paradigm [4]. The effect of methylphenidate depended on participants' trait impul-sivity, a measure previously associated with drug-induced dopamine release and D2/D3 (auto-)receptor availability [24]. Relative to placebo, methylphenidate increased the avoidance of effortful task-switching to a greater degree in more impulsive participants. These studies indi-cate that catecholamine interventions might have varying effect on motivated cognition across different individuals, likely as a function of baseline levels of dopamine function [25].

While prior work established a link between catecholamines and cognitive demand avoid-ance, it remains unclear whether catecholamine manipulation influences the value of cognitive control [4]. Here we employed a cognitive effort discounting task (COGED) that enabled us to explicitly quantify the value of cognitive control [16] and its modulation by a catecholamine challenge. The COGED paradigm consists of 2 phases: an effort execution phase, during which participants complete multiple levels of the demanding N-back task (levels 1–4 back) and an effort discounting task, during which participants choose between repeating a more demand-ing level for more money, or the 1-back for less money. Unlike the demand avoidance para-digm, choices are separated in time from performing the effortful task; as such, choices do not reflect learning of effort costs.

## 1.2 Tyrosine intervention in older adults

A second key way in which we go beyond prior studies is that we administer a catecholamine precursor (i.e. tyrosine) instead of a catecholamine transporter blocker (i.e. methylphenidate). Tyrosine is a precursor of dopamine and noradrenaline and the administration of tyrosine stimulates synthesis and release of catecholamines [26–30]. The main source of tyrosine is pro-tein-rich food, but tyrosine has also been administered selectively as an over-the-counter food supplement for study purposes and has been shown to alter cognition [31]. In young adults, tyrosine administration has been shown to improve cognitive control functions that are com-monly associated with catecholamine transmission, such as working memory, response inhibi-tion, and task switching [31,32].

In the present study, we administered tyrosine to older adults, aged 60–75, for the following 2 reasons: i) Healthy aging has been reported to be accompanied by a decline in dopamine transmission [33], making older adults perhaps more sensitive to tyrosine administration. A recent meta-analysis revealed lower prefrontal and striatal D1 and D2 receptor densities and striatal dopamine transporters with increasing age [34]. Diminished dopamine function is sup-ported by evidence of reduced reward responsivity in elderly, evidenced by impaired reward learning, attenuated BOLD signal in the ventral striatum in response to reward, and less risky choices in gain trials [35–37]. ii) The well-established decline in cognitive functioning with advanced age [38], has often been attributed to diminished cognitive control capacity, but may partly reflect motivational rather than capacity constraints. Thus, using the COGED paradigm, older adults have been shown to be *less motivated* to engage in effortful cognition [16]. Given that older adults are thought to exhibit diminished catecholamine function and they are less motivated to engage in cognitive effort, we speculate that lower catecholamine transmission contributes to reduced motivation for control. Following empirical and theoretical work on dopamine's role in cost-benefit analysis of cognitive actions, we hypothesized that augmenting catecholamine tone by the administration of the catecholamine precursor tyrosine can restore motivation for cognitive effort in older adults.

The effects of tyrosine administration have been shown to depend on the baseline state of the system. For example, tyrosine was shown to be particularly effective in enhancing cognitive control when the catecholamine metabolism was enhanced by acute stress or high cognitive demand, while having no or disruptive effects in other conditions where the need for

catecholamine transmission is lower [31,39]. Higher doses of tyrosine have been shown to increase plasma tyrosine concentrations to a greater degree in older than younger adults [40], and have been associated with poorer N-back performance than lower tyrosine doses [40]. Furthermore, tyrosine was recently found to reduce proactive response inhibition as a function of age [41]. Although we do not have direct measures of baseline catecholamine function in our sample, we explored in supplemental analyses whether the effects of tyrosine depended on two commonly used proxy measures of baseline dopamine levels: trait impulsivity and working memory capacity [see also 4,42]: trait impulsivity scores for their association with dopamine (auto)receptor availability and striatal dopamine release [24], as well as working memory span, associated with striatal dopamine synthesis capacity [43,44]. Previously, we have shown that trait impulsivity predicts the degree to which methylphenidate modulates demand avoidance [4]. Thus, while our study was set up to assess the hypothesis that tyrosine administration would increase the value of cognitive control, we also explored whether tyrosine altered the value of cognitive control in a manner that depended on either of two dopamine proxy measures.

## 2 Methods

### 2.1 Participants

Exclusion criteria for this study were a history of clinically-significant psychiatric, neurological or cardiovascular disorder, abuse of drugs or alcohol, abnormal blood pressure (< 90/60mmHg or > 160/90 mmHg), medication use that can interfere with tyrosine, blindness or colorblindness, smoking more than 1 pack of cigarettes per week, or contra-indications for MRI. For a complete list of exclusion criteria, see S1 File.

After a screening session, thirty-three healthy, right-handed adults were initially included for participation. However, four additional participants were excluded or decided to discontinue during or after the first experimental session, due to blood pressure exceeding our inclusion criteria (n = 1) and fMRI-intolerance (anxiety: n = 1; nausea: n = 1, headache: n = 1), leaving a sample of 29 participants who completed both experimental sessions (age: $M$ = 66.7, range = 61–71, 16 men). Our paradigm consists of two phases (see §2.4): an effort execution N-back task and a cognitive effort discounting (COGED) task. The COGED task is of primary interest to our research question and we have 29 complete datasets available. Due to technical, back-up problems, we have 26 complete datasets of the effort execution N-back task (day 1 = 27; day 2 = 28), even though all 29 participants completed this task. As our primary research question regards the COGED task, we report questionnaire and neuropsychological assessment data for the complete sample (n = 29) in Table 1. All procedures were in accordance with the local ethical guidelines approved by the local ethics committee ("Commissie Mensgebonden Onderzoek regio Arnhem–Nijmegen"; CMO protocol NL49758.091.14) and in line with the Helsinki Declaration of 1975.

### 2.2 Procedure

A within-subjects, placebo-controlled, double-blind, cross-over design was employed. Participants visited the institute three times: once for a screening and twice for experimental sessions of around 4.5 hours (Fig 1).

The screening session included reviewing additional information about the study and signing informed consent forms and was mainly designed to check for medical exclusion criteria (see S1 File). To assess specific exclusion criteria, we administered the Hospital Anxiety and Depression scale (HADS, [45]), Mental State Examination (MMSE, [46]) and the Dutch reading test for an estimate of verbal intelligence (NLV, [47]). In addition, participants' trait

**Table 1. Data from questionnaires, neuropsychological assessment (NPA), mood, blood pressure and urine metabolites.**

| | Measure | Screening | Placebo | Tyrosine | Drug effect |
|---|---|---|---|---|---|
| **Exclusion criteria** | HADS | 3.7 (2.6) | N/A | N/A | N/A |
| | MMSE | 29.1 (1.3) | N/A | N/A | N/A |
| | NLV—IQ estimate | 114.9 (8.6) | N/A | N/A | N/A |
| **Effort questionnaire** | NCS | 50.7 (11.8) | N/A | N/A | N/A |
| **Dopamine proxies** | BIS-11 | 58.2 (6.3) | N/A | N/A | N/A |
| | Digit span | N/A | 13.3 (3.3)* | 12.7 (3.7) | t(27) = 1.5, p = 0.145 |
| **General neuropsychological assessment** | Story recall—immed | N/A | 9.8 (2.8) | 10.6 (3.2) | t(28) = -1.2, p = 0.254 |
| | Story recall—delay | | 9.3 (2.7) | 9.7 (3.0) | t(28) = -0.6, p = 0.535 |
| | Stroop effect (sec) | N/A | 84.3 (48.6)* | 87.7 (73.7)** | t(26) = -0.2, p = 0.815 |
| | Verbal fluency, total | N/A | 46.3 (9.4) | 44.0 (10.1)* | t(27) = 1.2, p = 0.246 |
| | Box completion, min | N/A | 90.0 (33.7)* | 82.5 (19.4) | t(27) = 1.2, p = 0.248 |
| | Digit cancellation, min | N/A | 246.8 (31.4)** | 250.8 (38.9)* | t(25) = -0.8, p = 0.433 |
| **Mood** *(T1-T0)* | Calmness | N/A | -0.5 (1.7) | -0.9 (1.9)* | t(27) = 0.7, p = 0.485 |
| | Contentedness | N/A | -0.6 (1.4) | -0.7 (1.3)* | t(27) = 0.1, p = 0.888 |
| | Alertness | N/A | -0.2 (1.1) | -0.1 (1.0)* | t(27) = -0.6, p = 0.584 |
| | Total | N/A | -0.4 (1.0) | -0.4 (1.0)* | t(27) = -0.1, p = 0.906 |
| **Blood pressure** *(T1-T0)* | Systolic | N/A | 4.1 (8.6) | -0.6 (6.7) | **t(28) = 2.1, p = 0.041** |
| | Diastolic | N/A | -2.7 (5.1) | -2.3 (4.4) | t(28) = -0.3, p = 0.740 |
| | Heart rate | N/A | 0.1 (5.1) | -0.2 (3.4) | t(28) = 0.3, p = 0.768 |
| **Metabolites in urine** *(T1-T0)* | DOPAC | N/A | -0.03 (0.3) | 0.2 (0.4) | **t(28) = -3.0, p = 0.006** |
| | HVA | N/A | 0.6 (0.8) | 0.5 (0.7) | t(28) = 0.9, p = 0.370 |
| | VMA | N/A | 0.3 (0.2) | 0.3 (0.2) | **t(28) = 2.1, p = 0.048** |
| | MOPEG | N/A | 0.2 (0.1) | 0.2 (0.2) | t(28) = 1.5, p = 0.157 |

Measures are acquired during screening or testing days (see §2.2). Data represent mean (standard deviation) and when administered in both experimental sessions, results of paired-sample t-tests are presented to assess intervention effects. For the NLV-score and BIS-11 score, data points of 3 and 1 participant(s) were missing, respectively.

impulsivity (BIS-11, [48]) and Need for Cognition (NCS scale, [49,50]) were assessed because we had explicit, though exploratory, questions, related to the COGED paradigm (see §2.4). Scores of these self-report questionnaires are presented in Table 1. Included participants were also familiarized during the screening session with the cognitive test battery that was administered during the subsequent experimental sessions. This familiarization consisted of practice of a response inhibition task [41], a working memory task [cf. 51,52] and the N-back task [see §2.4, based on 16]. Afterwards, participants were guided to the fMRI facility and their weight was assessed for adequate dosage calculation (see §2.3).

The two experimental sessions were identical, except that participants received placebo on one day and tyrosine on the other (counterbalanced across participants). Participants were asked to come to the lab in the morning (at 8 am or 10 am) after overnight fasting: they refrained from eating, drinking except from water, and taking any medication after 10 pm of the previous day. The overnight fast reduces variability in plasma large neutral amino acid levels between participants caused by the previous meal [53]. A similar fasting procedure has been adopted in other research using tyrosine supplementation [54–58]. Sessions started approximately at the same time of the day (maximal deviation was 90 minutes), with an interval of one week to a max of 17 weeks between testing days. After informed consent, participants practiced the response-inhibition task [41], and right after drug administration (see

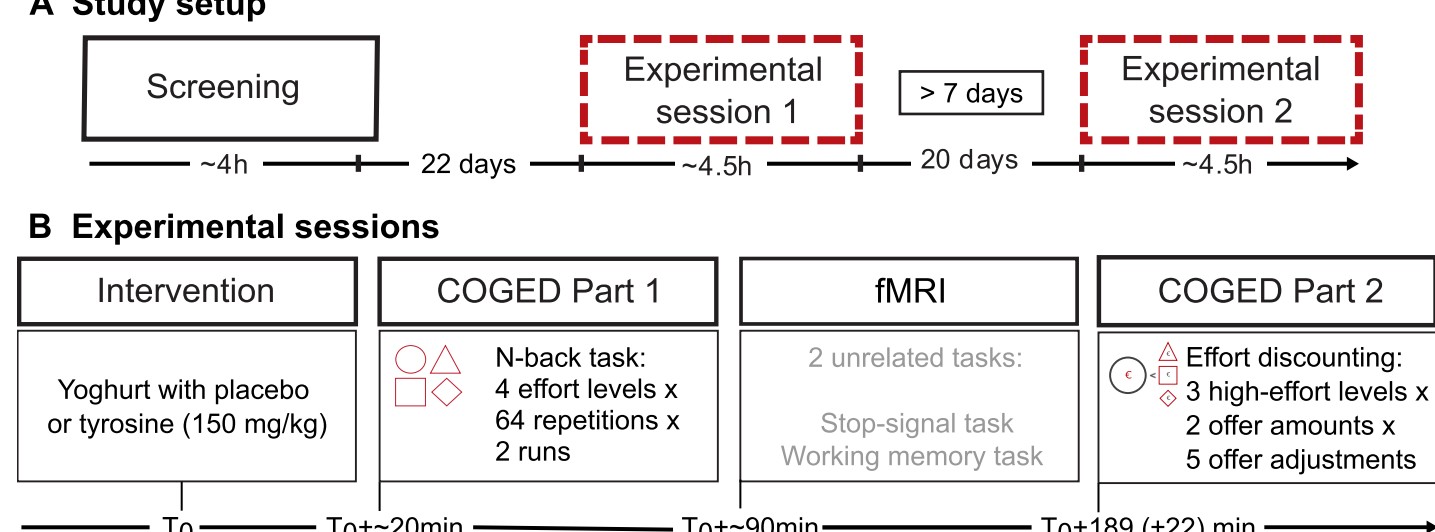

**Fig 1.** Schema of study setup **(A)** and experimental sessions **(B)**. **A** An initial screening was followed by two identical (except for placebo versus tyrosine intervention) experimental sessions. Duration between screening and session 1 was on average 22 days, between the two experimental sessions on average 20 days. To prevent any carry-over effects of pharmacological interventions, the experimental sessions were separated by at least 7 days. **B** During the experimental sessions, participants received placebo or tyrosine and conducted a test battery (see §2.2). The effort discounting choice task was administered after an fMRI session and took place around 3 hours after the yoghurt ingestion. Plasma levels of tyrosine have been shown to peak 90–120 minutes after the oral tyrosine administration.

§2.3), each level of the N-back task was rehearsed followed by the practice of another working memory task [cf. 52]. The cognitive test battery consisted in total of 3 paradigms (Fig 1). The order of practice and paradigms was constant across sessions and participants, such that the effort execution N-back task was always administered soon after drug intake (~+ 20min). To isolate effects on choice from effects on execution, the effort execution (i.e. N-back) task was timed to immediately follow ingestion of the intervention, so that tyrosine was highly unlikely to have taken effect during task execution, given its delay in reaching peak concentrations (±2 hours, see §2.3). Then, after a break of 90 minutes, the response inhibition task [41]) and working memory task [cf. 52] were administered during fMRI. After fMRI (duration ~90 minutes), the COGED task was administered together with a N-back redo which is based on participants' choices (see §2.4). The delay between tyrosine administration and the COGED task (described in §2.4) was on average 189 (+/- 22) minutes, while plasma tyrosine levels have been measured to remain elevated up to 8hrs [see §2.3, 27].

After task completion, we administered different neuropsychological tests, including immediate and delayed story recall [59], digit span forward and backward [60], Stroop cards [61], verbal fluency [62], box completion [38] and number cancellation [63]. Summary scores are presented in Table 1. For safety reasons, blood pressure and heart rate were measured three times throughout the days (start of testing day, before task battery, after task battery). At the same time points, participants' mood was assessed using the Bond and Lader Visual Analogue Scales (calmness, contentedness, alertness; [64]). For exploratory purposes, assessing tyrosine's effect on dopamine metabolites, urine was collected on both testing sessions off drug (i.e. before drug administration) and around the peak of tyrosine concentration (i.e. right after the fMRI part). Intervention effects on mood, blood pressure and urine data (all T1-T0 due to peak level of intervention) are reported in Table 1. S2 File reports mood and blood pressure data for T2-T0. To probe awareness of the drug manipulation, participants reported after the second testing day their belief about the order of placebo and tyrosine sessions.

## 2.3 Tyrosine administration

Participants received tyrosine on one and a placebo substance on the other day, both adjusted to body weight as determined during the screening session (see §2.2). Following multiple previous studies in young volunteers (e.g. [54,58]; but see e.g. [65]), we administered 150 mg/kg L-tyrosine powder (BulkpowdersTM, Sports Supplements Ltd. Colchester, Essex, United Kingdom). The placebo product was a mixture of 54 mg/kg dextrine-maltose (Fantomalt by Nutricia) with 110 mg/kg cornstarch (ratio Fantomalt/cornstarch = ~½). The ratio of Fantomalt to cornstarch was adjusted to ensure that placebo and tyrosine mixture have an equal energy level, similar structure and aftertaste. Tyrosine and placebo powders were mixed with 200g of banana-flavored yoghurt (Arla Foods Nederland, Nijkerk, The Netherlands) to ensure comfortable ingestion. In a formal blinded sensory experiment, a specialized dietician from the Division of Human Nutrition of Wageningen University (E. Siebelink) confirmed equal taste experience of the two mixtures. Weighting of the doses, preparing and coding the samples were performed by a staff member not involved in the study, thus the order of administration was double-blind.

Tyrosine is a catecholamine precursor: when tyrosine enters the brain via the blood-brain barrier, it is converted into levodopa through the rate-limiting enzyme tyrosine-hydroxylase (TH; [66]) and then further converted into dopamine through the enzyme aromatic l-amino acid decarboxylase (AADC). In turn, dopamine can be converted into noradrenaline through the enzyme dopamine beta-hydroxylase (DBH; [31,67]). The oral administration of tyrosine significantly enhances central catecholamine synthesis in rodents [26,29,30,68–70] and humans [28]. Plasma concentrations peak ~2h after administration and remain significantly elevated up to 8h [27]. The administration of 150 mg/kg body weight tyrosine has been shown to significantly increase plasma tyrosine concentrations also in older adults, peaking at 90 minutes and remaining elevated till at least 240 minutes after drug intake [40]. To test participants at maximal plasma levels, participants underwent the cognitive test battery starting ~90 minutes after drug intake. The delay between tyrosine administration and the COGED task (described in §2.4), the paradigm of primary interest for our research question, was on average 189 (+/- 22) minutes.

## 2.4 Task design

The task design was, except for minor adaptations, identical to that described in [16]. Each experimental session consisted of an effort execution N-back phase (§2.4.1, Fig 2A), the cognitive effort discounting phase (COGED; §2.4.2, Fig 2B) and additional N-back rounds based on a random selection from among their choices in the discounting procedure. The entire protocol was programmed and administered using Psychophysics toolbox [71,72] in MATLAB.

**2.4.1 Effort execution: N-back task.** Participants completed the N-back task three times: a longer version during the screening session and a shorter version during the experimental sessions. The tasks were administered in behavioral labs with participants sitting comfortably in front of the screen, hands located on the keyboard. Participants were instructed to complete a working memory task in which they are presented with series of letters in the center of the screen and that they need to respond by indicating whether each letter is either a target or non-target by keypress (Fig 2A). All versions start with easiest, 1-back level and increased block-wise to the highest, 4-back level. In the 1-back task, participants compared the current letter to the letter presented 1 position (i.e. screen) back and if the letter was identical, they pressed the target key; if not, they pressed the non-target key. For the 2-back task, a target was defined as identical letter presentation 2 screens back, etc.

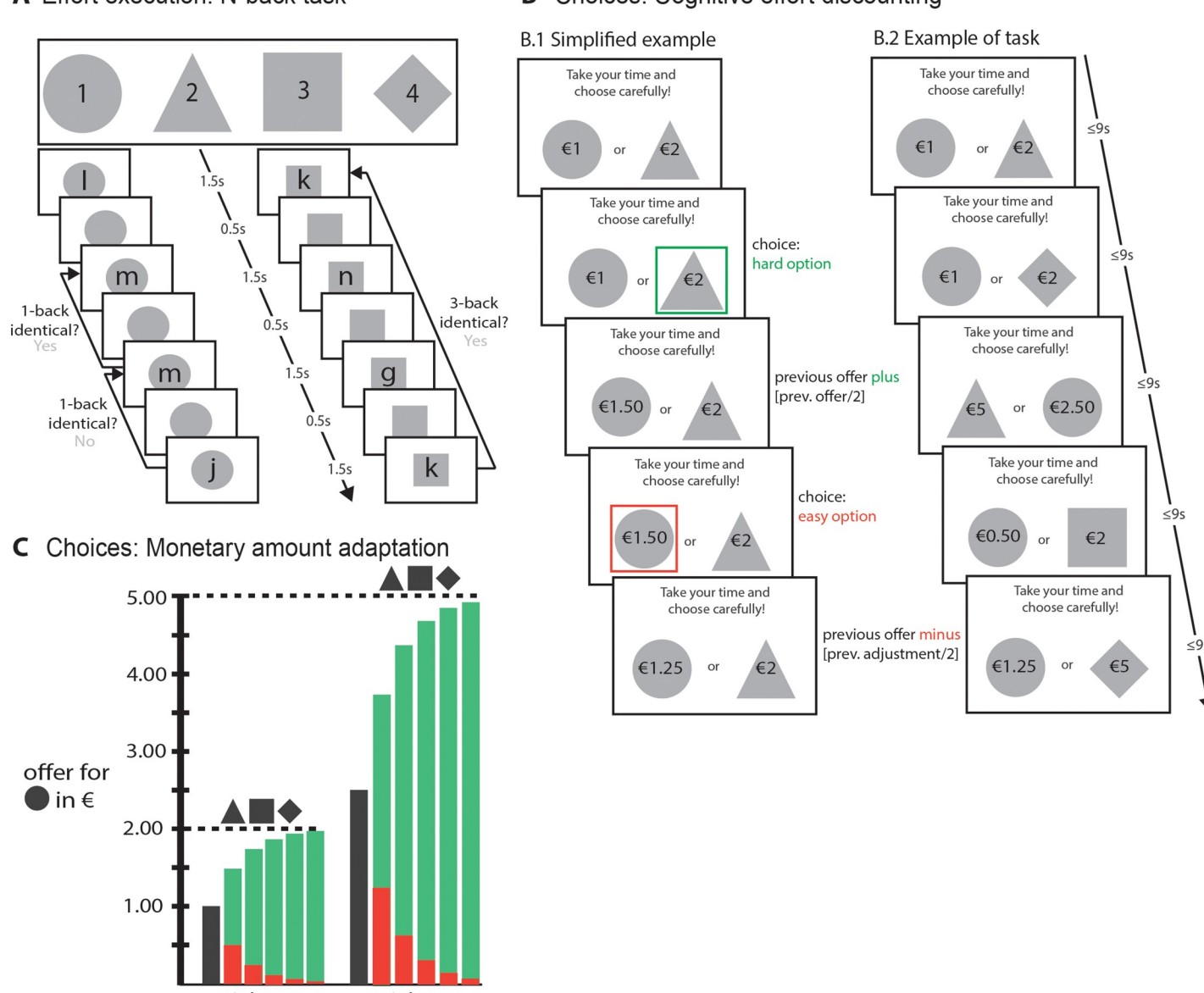

**Fig 2. The experimental paradigm was based on the procedure described in Westbrook et al., 2013. A.** The N-back task. Letters appeared serially on the screen for 1.5s, but disappeared after a response was given followed by an ITI of 0.5 s. Every trial had a total duration of 2 s. Participants indicated whether every letter as target or non-target by keypress. Target assignment depended on the N-back level represented by shapes. Levels were presented block-wise with increasing difficulty (circle up to diamond). **B.** The discounting task. Higher N-back levels (levels 2–4) were paired with the lowest level for varying amounts of money. **B.1.** A simplified illustration of one trial type: 1-back (circle) versus 2-back (triangle) in the low amount condition (harder task worth €2 instead of €5). The schema presents the monetary amount adjustment as a function of choice. The amount of the harder tasks is fixed, while the easier task varies. When the hard option is chosen, the amount offered for the easy task increases while it decreases when the easy choice is chosen. The amount adjustment reduces exponentially (by the power of 2), see C. **B.2.** In the real choice task, the trial types (level [3] x amount [2] x amount adjustment [5]) were randomized, resulting in a total of 30 choices. Choices were self-paced but had a maximal duration of 9 s. After the choice is made, a box is presented around the chosen option for 0.75 s. If no response was given, the message "Too slow!" was presented for 0.75 s. Minimal trial duration was set to 2 s. **C** Per harder task level (triangle, square, diamond), 5 choices were presented with respect to circle (1-back) for a varying amount offered for the easy task. Hard tasks were either fixed at €2 or €5. Red bars show the decrement if participants always chose the easy task, while the green bars show the increment if always the hard task was chosen. Participants' choices thus vary in this range. The adjusted amount decreased as a function of trial number of the specific pair. The subjective value is determined based on the last trials adapted following the last choice.

The practice phase during screening consisted of three runs for every load level, the experimental sessions had two runs. Each run comprises 64 items (consonants, 24-point Courier

New font, 16 targets, black font). Participants were instructed explicitly at the beginning of each new level, which level they were about to complete. In addition to this explicit information, dark grey shapes were presented in the background which participants could learn to associate with the different N-back levels, see Fig 2A. Beyond indicating the current task level and rules, the shapes had no other utility for performing the N-back task. Shapes had a diameter of 10 cm and were presented in the center of the lighter gray screen as the background of the letter stimuli on each trial. Participants had 1.5 s to respond to each item by button press, after which items were replaced by fixation cross. The inter-item interval was 0.5 s. Lures (items within N +/- 2, but not exactly N, positions after last presentation) were included in N-back stimulus lists to increase level difficulty: eight for N = 1, six for 2, five for 3, and three for N = 4. Participants were given feedback about run-wise performance ("% of targets" and "% of non-targets correct"). To motivate engagement, and to prevent participants from responding, e.g., "Non-target" at the expense of the "Target" score, participants were also given feedback of "Good job!" if both scores were above 50% or "Please try harder!" otherwise. Additionally, after each level of N-back experience, participants completed a self-report questionnaire reflecting on their task experience (see S3 File). Participants indicated on a Likert scale from 1 to 10 how difficult and effortful they perceived the task and, for higher levels, how effortful the level was compared with level 1.

**2.4.2 Choices: Cognitive effort discounting.** The discounting procedure was also administered in behavioral labs with task presentation on a pc and responses given on the keyboard. In the discounting procedure, on every trial, participants made choices between a higher N-back level (levels 2–4) for a fixed monetary amount (€2 or €5) and the 1-back task for a lower, variable amount (Fig 2B), analogous to adjusting-immediate-amount (AIA) procedures used in intertemporal and risky choice [73]. Participants were told that they could choose which N-back level they want to repeat for earning a monetary bonus and that one of their choices would be randomly selected and played out: they would repeat 1–10 runs of the N-back level that they selected and receive the monetary bonus attached to their choice. To reduce avoidance of mistakes rather than effort, we instructed participants that they would receive the bonus if they do their best and perform comparable to the practice round on the same day.

Choice options were presented on the left and right side of the screen. Levels were referred to by the same shapes that participants learned to associate with each N-back level during practice and effort execution on the same day (see §2.4.1). To minimize confusion about levels, participants also had access to a paper sheet reminding them of the relevant shape-level associations. The amount of the monetary bonus was presented in the center of the shape (36-point Courier New font, black font). Each of the 3 higher levels (N = 2–4) was paired with the easier 1-back level in two different amount categories: higher levels were either offered at €2 or €5. For the first paring, the amount offered for the easy task was half the amount offered for the harder task, thus €1 or €2.50, respectively. Depending on participants' choices, the amount offered for the easy task was adjusted (see Fig 2B.1): when participants chose the harder/high offer option, the amount offered for the easy task on the next pairing was increased; when the easier/low offer option was chosen, the amount offered on the next trial would decrease. The magnitude of amount adjustments was cut in half after each adjustment such that the offer for the easy task converged towards a point of indifference. Fig 2C presents the adjustment path for the easy offer when participants always select the easy (red bars) or hard (green bars) task. Note that this titration procedure implies that initial choices weight more heavily than later choices due to the decreasing amount of offer adjustments throughout the choice phase. Yet, from a previous study it seems that the applied titration procedure approximates an indifference point quite reliably [74]. Moreover, each level-specific subjective value is based on two

independent choice runs (low and high offer amount), which reduces the impact of initial choice "mistakes" in one run on overall subjective value estimation.

The choice task comprised a total of 30 choices (3 levels * 2 amounts * 5 amount adjustments). Trial types and offer orders were randomized. Choices were self-paced but have a maximal duration of 9 seconds. The text "Take your time and choose carefully" was presented at the top of the screen during all choices. If no choice was made within 9 sec the text "Too slow!" was presented.

The indifference point reflects the monetary amount offered for the easy task on the last trial corrected for the last choice and was assessed for each level (levels 2–4) per amount condition (€2 and €5). "Subjective value" (SV) hereafter refers to indifference points divided by the amount category (€2 or €5), such that all numbers ranged from 0 to 1 for both the low and high amount offers. A SV of 0.8 means that a participant is equally likely to choose one or the other option (i.e. indifferent) when the easier task is worth 80% of the amount offered for the harder task. A lower SV thus indicates that a participant chooses to receive less money but increases the likelihood to redo an easier task. After the choices paradigm, all participants completed their randomly selected choice exactly four more times and were paid the associated amount for each repetition.

## 2.5 Questionnaires and digit span

A series of questionnaires and neuropsychological tests were completed by participants during the screening and experimental sessions. Trait impulsivity, digit span and Need for Cognition Scale were included in our secondary, exploratory analyses and will be described in more detail below. Scores on other acquired measures are presented in Table 1.

**2.5.1 Trait impulsivity.** The Barratt Impulsiveness Scale (BIS-11; [48]) was administered to assess participants' degree of trait impulsivity. The scale is a self-report questionnaire, consisting of 30 statements that participants rate on a 4-point Likert scale ("never" to "almost always"). Examples are "I buy things on impulse" or "I am future oriented". Scores on this questionnaire can range from 30 to 120. The total Barratt score has been found to be associated with reduced dopamine D2/D3 receptor availability in the midbrain, and enhanced dopamine release in the striatum [24,75] and has been shown to predict effects of MPH on learning [76]. This measure served as a putative proxy of baseline dopamine function in the exploratory analyses (see §2.6).

**2.5.2 Digit span.** Baseline working memory capacity was assessed with the digit span [60] at the end of both experimental sessions. The digit span consists of two parts: forward and backward digit span. In the first part, participants' task was to repeat series of numbers that are presented via headphones in the same order as presented (forward). Series start with three numbers and increase up to 9 numbers. Participants complete two trials for each span and their score is identical to the maximum of digits repeated without any error in one of the two trials. The second part is almost identical, except that participants have to repeat the span backwards, beginning with the last digit of the span. The lowest span contains two, and the highest eight digits. Here too, the score is equal to the maximum of digits repeated correctly. Forward and backward scores are added to obtain a total score, such that scores can range from 0 to 17. In the absence of tyrosine effect on this measure, as in earlier studies [77], the average total digit span across two days was selected, because it was thought to provide a more reliable estimate of working memory capacity. The total scores were averaged across the assessments and used as putative proxy of baseline dopamine function in exploratory analyses (see §2.6).

**2.5.3 Need for cognition.** The self-report Need for Cognition Scale [49,50] was administered to investigate participants' tendency (trait) to engage in effortful tasks. The scale consists

of 18 statements, which participants rate on a 5-point Likert scale ("extremely uncharacteristic of me" to "extremely characteristic of me"). Example statements include "I prefer complex to simple problems" or "I only think as hard as I have to". Scores range from 18 to 90.

## 2.6 Statistical analysis

**2.6.1 Effort execution: N-back task.** The N-back task was used to expose participants to different levels of working memory load. To assess whether performance on the N-back task was sensitive to the load manipulation, we analyzed performance measures: response times and signal detection d' as a measure of sensitivity to targets corrected for the propensity to make a target response (false alarms).

Note that the N-back task was conducted right after drug intake and therefore we did not predict any intervention effects. However, to rule out that N-back performance differed between the experimental session, we assessed drug effects on response times and d' also as a function of N-back levels. The data were analyzed with a linear mixed-effects model approach using the lme4 package in R [78,79]. This allowed us to account for within-subject variability in addition to between-subject variability. Drug (tyrosine versus placebo) and level (levels 1–4) were within-subject factors. The model included all main effects and interactions and a full random-effects structure [80]. To determine p-values, we computed Type 3 conditional F tests with Kenward-Roger approximation for degrees of freedom as implemented in the mixed function of the afex package [81], which in turn calls the function KRmodcomp of the package pbkrtest [82]. Effects were considered statistically significant if the p-value was smaller than 0.05.

Given impulsivity-dependent effects of tyrosine on SV (§3.3) and to assess whether choice effects could be a consequence of (unexpected) effects of tyrosine on performance, we extended the performance models post-hoc to include participants' trait impulsivity scores (BIS-11) and working memory capacity (digit span average) as (z-scored) between-subject factors. An overview of all performance models is presented in S4 File (Models 2.1–2.4).

**2.6.2 Choice task: COGED.** The experiment was set up to assess effects of tyrosine on the valuation of cognitive effort. We therefore estimated participants' subjective values for the three higher N-back levels in two amount conditions (€2 or €5) for the placebo and tyrosine sessions. Values range from 0 to 1 and represent the subjective value with respect to the 1-back. Drug (tyrosine vs. placebo), level (levels 2–4) and amount (€2 vs. €5) were within-subject factors. The procedure of model estimation and p-value extraction were identical with that described in §2.6.1. Relatedly, we explored whether tyrosine modulated the speed of choosing (i.e. median choice response time), by running a model with identical predictors as the choice model described here, but median response times as dependent variable.

In further exploratory analyses, we added participants' trait impulsivity (BIS-11) and working memory capacity (digit span average) as (z-scored) between-subject factors to the basic model. Due to missing data for one participant of the trait impulsivity measure, the sample for this analysis is 28. An overview of all choice models is presented in S4 File (Models 1.1 and 1.2). Note that Model 1.2 does not include in the random-effects term the factor 'offer amount' due to convergence-warnings. Nevertheless, statistics of the effects of interest as obtained with the complete model are presented for completeness in S9 File.

**2.6.3 Questionnaire data.** *Self-report N-back.* After each N-back level during effort execution (see §2.6.1), participants judged difficulty, effort and effort with respect to level 1 (for higher levels) using a 1–10 Likert scale (see S3 File). As for N-back performance and choice data, we analyzed whether the perceived difficulty and effort increased as a function of N-back level. To assess whether differences in perceived effort existed immediately after drug intake,

we analyzed these measures as a function of drug with three separate repeated-measures ANO-VAs (difficulty, effort, effort with respect to N = 1) in SPSS 23 (IBM Corp., Armonk, N.Y., USA).

*Need for Cognition.* We included the Need for Cognition Scale to assess whether we can replicate a (positive) relationship between SV as quantified with the COGED-task and Need for Cognition scores as reported in [16]). Thus, we ran another mixed-effects model in R with SV as dependent variable and the factors level and amount as within- and Need for Cognition scores as between-subjects predictors. Results are reported in S5 File.

**2.6.4 Control analyses.** We performed a number of control analyses using a model comparison approach (anova function in R) where we assessed whether the residual sum of squares was significantly reduced when adding any of the following, perhaps confounding, factors to the SV model: order of drug administration, gender, age, and NLV scores (as a measure of verbal intelligence). Furthermore, we added an additional control analysis to assess directly whether the drug effect of interest (§3.3; Fig 5) was altered when including the factor order in a statistical model (S6 File).

Given that N-back data is available for 26 instead of 29 participants, we repeated the choice analyses (reported in §2.6.2) for the smaller sample as an additional control analysis. We also assessed in this sample whether the inclusion of the drug-induced performance changes on the N-back task ($d'_{Tyr}$-$d'_{Pla}$ and $RT_{Tyr}$−$RT_{Pla}$) in the choice analyses (§2.6.2) still reveal significant modulations observed in 3.3 (Fig 5). Note that the N-back task was performed shortly after drug administration, but before tyrosine-levels are expected to peak [27,40].

# 3 Results

## 3.1 Effort execution: N-back task

Participants performed well on the N-back task, evidenced by an overall proportion of 0.83 and 0.84 correct responses on the placebo and tyrosine session, respectively. This corresponds to d'-values of 2.02 and 2.03. In line with earlier work [16], performance was sensitive to the load manipulation: d' decreased as a linear function of N-back levels (level effect: $F(1, 25) = 129.45$, $p < 0.001$; Fig 3A), while response times increased (level effect: $F(1, 25) = 20.99$, $p < 0.001$; Fig 3B). As expected given our design, tyrosine had no main effects on performance, as assessed by d' (drug effect: $F(1,25) < 0.01$, $p = 0.978$) and response times (drug effect: $F(1, 25) = 0.94$, $p = 0.342$), or interactions with N-back level (for d': drug x level interaction: $F(1, 25) = 0.29$, $p = 0.596$; for RTs: drug x level interaction: $F(1, 25) = 2.86$, $p = 0.103$). This lack of drug effects is not surprising, because participants performed the N-back task immediately after tyrosine administration, so brain tyrosine levels were unlikely to have risen at the time of the N-back task performance. Average performance data (d' and RTs) as a function of level and drug are presented in Table 2. For a complete list of statistical effects, see S7 File.

## 3.2 Choices: Cognitive effort discounting

As expected, participants' COGED choices indicate a decline in subjective value (SV) when N-back levels increased (level: $F(1,28) = 54.08$, $p < 0.001$), indicating that effort costs increased with working memory load. Surprisingly, when higher amounts were offered in the discounting task (i.e. €5 instead of €2), participants' SV of the N-back task was slightly lower (amount: $F(1, 28) = 4.77$, $p = 0.037$). While the load effect is in line with earlier reports using this task, the latter amount effect was unexpected given that prior work has shown shallower discounting for larger rewards in both cognitive effort [16] and delay discounting [83]. In addition to these manipulation checks for SV, we analyzed the speed by which choices were made. Participants chose faster when more money was at stake (i.e. €5 versus €2; amount for RTs: $F(1, 28) = 4.1$,

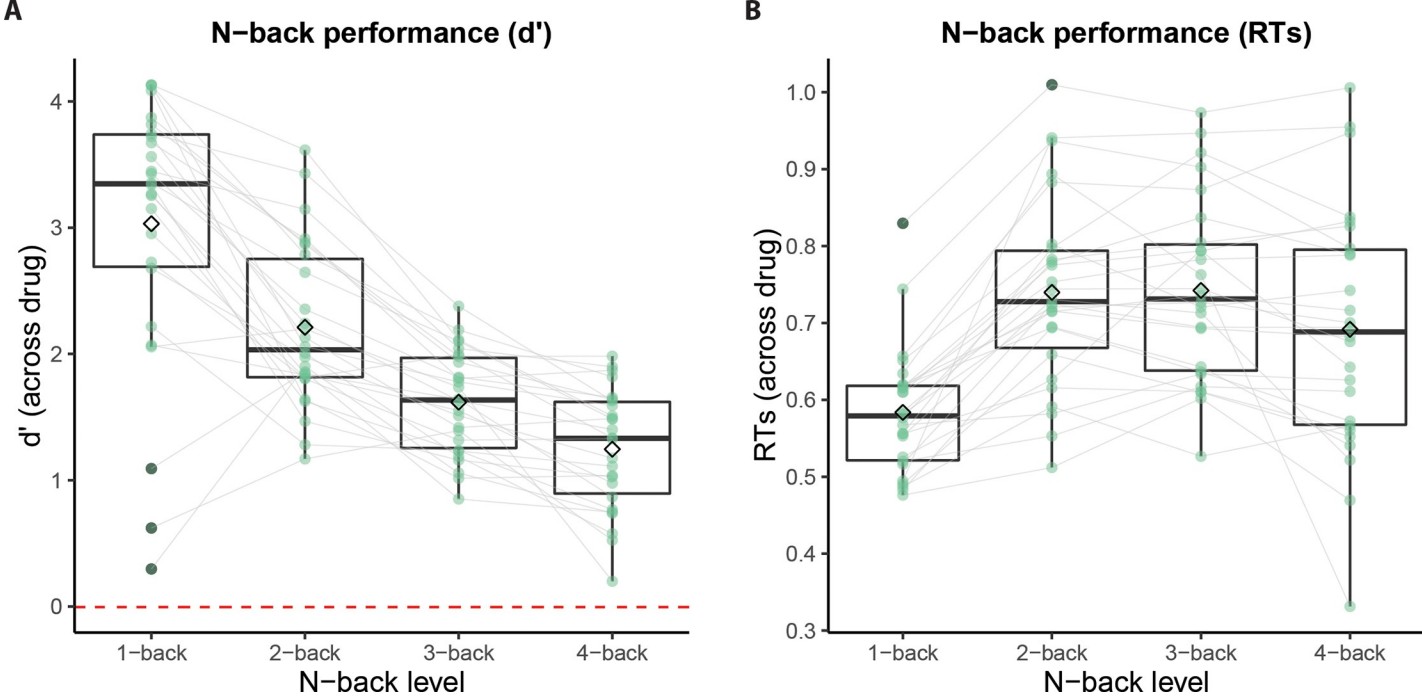

**Fig 3. Performance on the N-back task as a function of working memory load (i.e. levels) across drug. A** $d'$, the estimate of participants' sensitivity to targets corrected for the propensity to make a target response (false alarms), decreased as a function of N-back levels. **B** Response times increased as a function of N-back level, but showed an inverted U-shape, in line with earlier report using this task [16]. In both graphs, the horizontal lines in the boxplots represent the median, the diamond represents the mean and green dots are the mean per subject. Average $d'$ measures contain outliers in level 1 (see dots with gray edge) because on one of the testing days these participants initially swapped keys for target vs. non-target (only in level 1) and had negative $d'$ for one session.

p = 0.027). Response times numerically, though not significantly, decreased as a function of N-back level (level for RTs: $F(1, 28) = 4.1$, $p = 0.053$).

Critically, we hypothesized that the administration of tyrosine raises participants' motivation for cognitive control, evidenced by higher SV of the N-back task compared with the placebo session. In contrast to this hypothesis, tyrosine did not significantly increase overall valuation of the N-back task (drug: $F(1, 28) = 0.15$, $p = 0.699$), and there was no interaction with level (drug x level: $F(1, 228) = 0.01$, $p = 0.912$; Fig 4A). Choice response time analysis revealed that tyrosine numerically increased overall response times, but the effect was not statistically significant (drug: $F(1, 28) = 3.8$, $p = 0.060$). Average SV as a function of level and drug

**Table 2. Group average (and standard deviation) of performance data (d' and RT) on the N-back task and subjective value of the cognitive effort discounting task.** Note that subjective values of higher N-back levels are all relative to level 1, thus no values are available for level 1.

|  | Level 1 | Level 2 | Level 3 | Level 4 | Average |
|---|---|---|---|---|---|
| **d'** | 3.03 (1.49) | 2.21 (0.70) | 1.62 (0.49) | 1.25 (0.52) | 2.03 (1.12) |
| placebo | 2.92 (1.68) | 2.25 (0.73) | 1.68 (0.49) | 1.25 (0.47) | 2.03 (1.15) |
| tyrosine | 3.14 (1.30) | 2.17 (0.69) | 1.56 (0.50) | 1.24 (0.57) | 2.03 (1.09) |
| **RT** | 0.59 (0.09) | 0.74 (0.13) | 0.74 (0.13) | 0.70 (0.16) | 0.69 (0.14) |
| placebo | 0.58 (0.18) | 0.74 (0.25) | 0.75 (0.25) | 0.72 (0.27) | 0.70 (0.25) |
| tyrosine | 0.60 (0.19) | 0.74 (0.24) | 0.73 (0.24) | 0.68 (0.26) | 0.68 (0.24) |
| **Subjective value** | N/A | 0.63 (0.36) | 0.28 (0.35) | 0.19 (0.28) | 0.37 (0.38) |
| placebo | N/A | 0.65 (0.37) | 0.28 (0.37) | 0.20 (0.30) | 0.38 (0.40) |
| tyrosine | N/A | 0.62 (0.34) | 0.27 (0.33) | 0.18 (0.26) | 0.36 (0.36) |

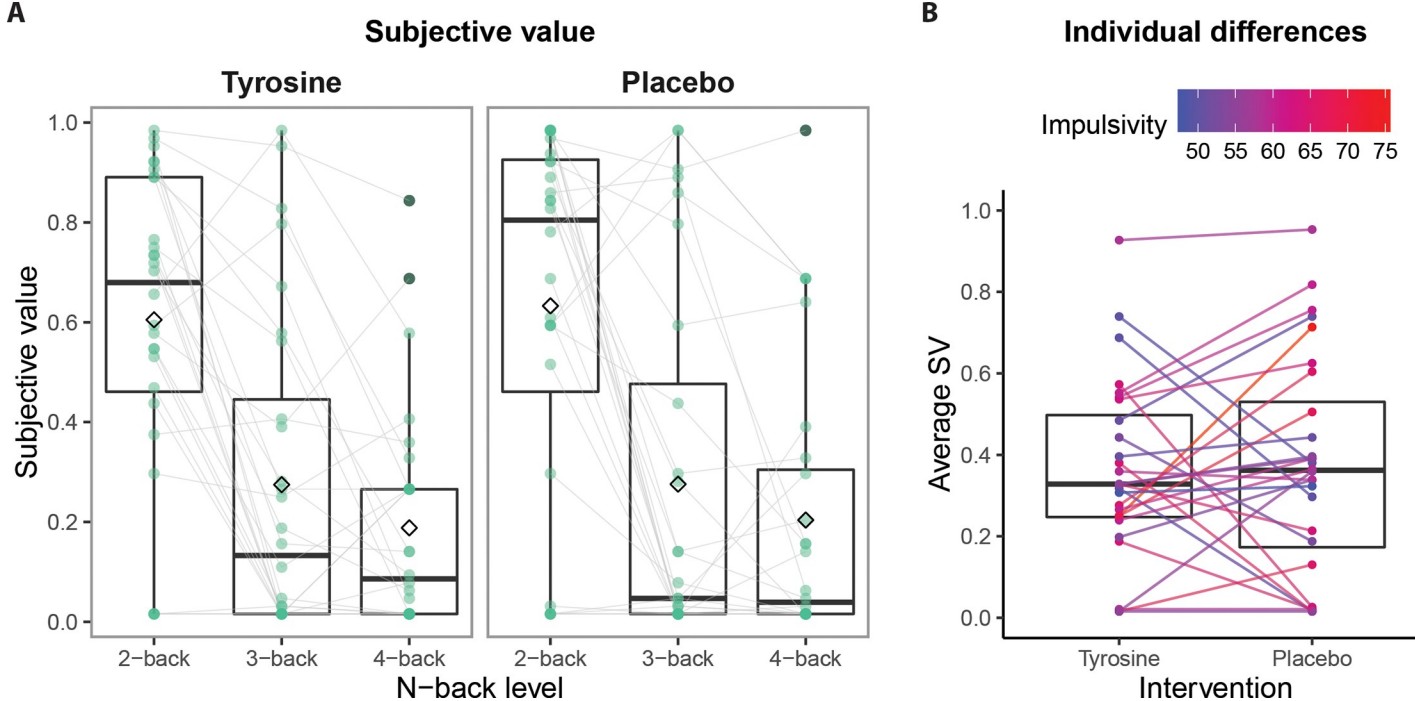

**Fig 4. Subjective value as measured by the cognitive effort discounting task. A** Subjective value for tyrosine and placebo sessions as a function of level. The horizontal lines in the boxplots represent the median, the diamond represents the mean, and green dots are the average subjective value per subject. Both sessions show that subjective value decreased with increasing working memory load. However, in contrast to our prediction, subjective value did not differ between the interventions. Individual outliers are marked with a gray edge. **B** Subjective value averaged across levels as a function of drug. The horizontal line in the boxplots represent the median. Lines show the change in subjective value per subject, color-coded for their trait impulsivity score, as a function of drug.

are presented in Table 2. For a complete list of statistical effects, see S8 File. S10 File shows the titration procedure (i.e. choice-dependent offer adjustments of the low-effort task) for individual and group data, similar to the description in Fig 2B.

## 3.3 Individual differences: Proxy measures of dopamine

Following earlier work indicating that catecholaminergic interventions depend on dopamine baseline levels [1] and the recent study showing that motivation for cognitive control depended on participants' trait impulsivity scores [4], we explored whether tyrosine effects on SV varied as a function of trait impulsivity (BIS-11) and working memory scores (digit span). As in our recent methylphenidate study, tyrosine effects on SV depended on participants' trait impulsivity scores, evidenced by two correlations of medium effect sizes: Tyrosine administration resulted in steeper SV discounting (i.e. higher cost) as a function of N-back levels in more relative to less impulsive participants (r = -0.37; drug x impulsivity x level: $F_{(1, 25)}$ = 5.01, p = 0.034; Fig 5). In addition to this level-dependent effect, tyrosine tended to also decrease the overall subjective value (i.e. irrespective of level) as a function of trait impulsivity (r = -0.33; drug x impulsivity: $F_{(1, 25)}$ = 4.19, p = 0.051; Fig 4B). Task effects did not significantly vary as a function of working memory capacity (drug x digit span: $F_{(1, 25)}$ = 1.29, p = 0.268; drug x digit span x level: $F_{(1, 25)}$ = 1.03, p = 0.320). A complete list of statistical effects is presented in S8 File.

Although we considered it unlikely that tyrosine could have altered N-back performance, given the timing of the intervention, we tested this assumption by adding impulsivity (and digit span) scores as covariates to the N-back models. This analysis also allowed us to assess

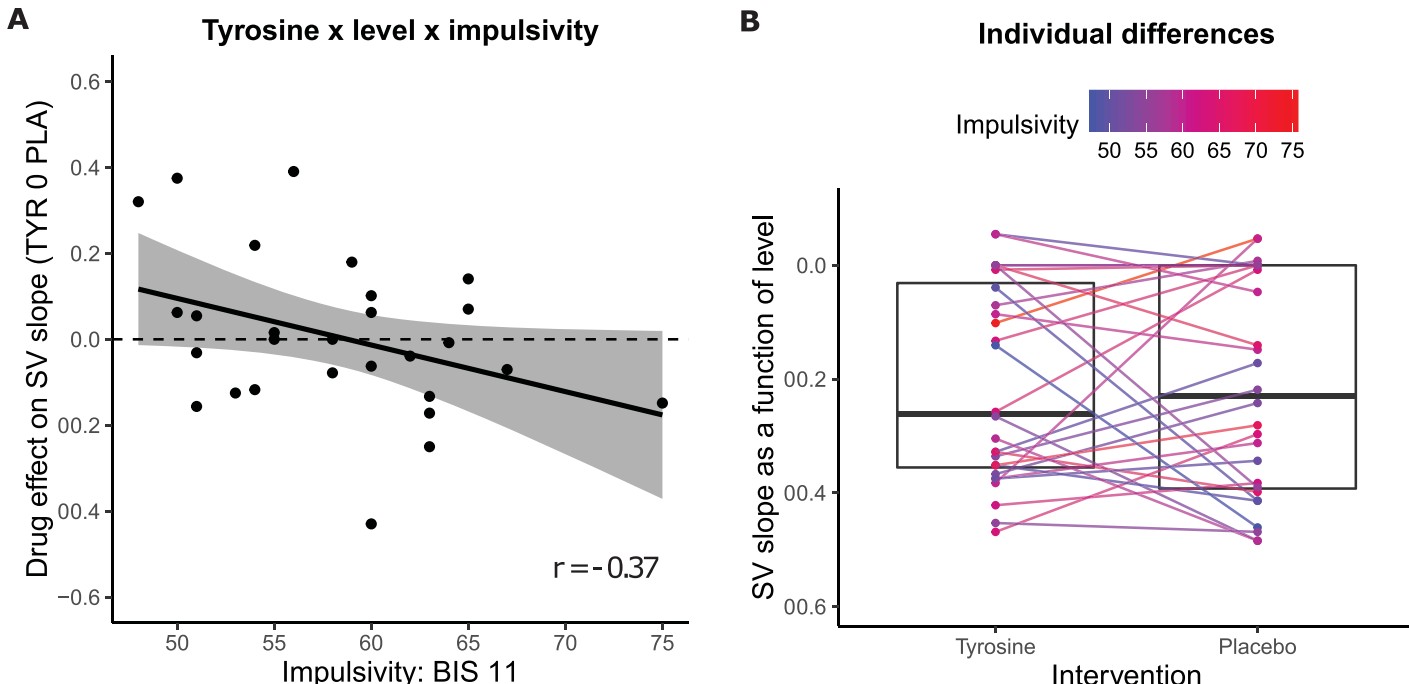

**Fig 5. Tyrosine-effects on subjective value vary as a function of participants' trait impulsivity (BIS-11) scores. A** Black dots represent per participant the difference scores (tyrosine minus placebo) of the slope of the subjective value as a function of N-back levels. Negative scores reflect more pronounced discounting (i.e. steeper subjective value slope) on tyrosine compared with placebo. The black line represents conditional means given the linear model used and shaded area represents the 95% confidence interval. The correlation between impulsivity scores and subjective value slope on placebo is r = 0.38, suggesting that more impulsive participants have a shallower reduction in SV as a function of increasing demand (i.e. N-back level). The correlation on tyrosine is r = 0.03. **B** Subjective value slopes as a function of drug. The horizontal line in the boxplots represent the median. Lines show the change in subjective value slope per subject, colour-coded for their trait impulsivity score, as a function of drug. Negative values on the y-axis indicate that SV decreases as a function of higher N-back levels: a steeper slope represents steeper discounting with respect to level 1.

whether this impulsivity-dependent effect of tyrosine on effort discounting is an indirect consequence of an impulsivity-dependent effect on performance (e.g. reflecting error avoidance). As expected, given that the N-back task was performed before tyrosine levels were peaking, we did not observe such impulsivity-dependent tyrosine effects for d' (drug x impulsivity: $F_{(1, 22)}$ = 0.03, p = 0.863; drug x impulsivity x level: $F_{(1, 22)}$ = 1.02, p = 0.323). Tyrosine also did not alter overall response times as a function of impulsivity (drug x impulsivity: $F_{(1, 22)}$ = 2.25, p = 0.148). However, surprisingly, tyrosine administration attenuated level-related slowing to a greater degree in more impulsive participants (drug x impulsivity x level: $F_{(1, 22)}$ = 4.86, p = 0.038). For a complete list of statistical effects, see S7 File. To further exclude that the tyrosine-induced reduction in SV was driven by failure-avoidance in more impulsive participants, we assessed whether we could replicate the (significant) modulation of SV by tyrosine and impulsivity when including tyrosine-induced performance changes in the SV model. Note that this control analysis is based on 25 instead of 29 datasets due to missing N-back (n = 3) and impulsivity (n = 1) data and might thus also suffer from a reduction in power to detect an effect. Despite the smaller sample and the inclusion of (drug-induced) d'-scores, we replicated the effect of interest (drug x level x impulsivity: $F_{(1, 19)}$ = 4.8, p = 0.041), suggesting that an indirect modulation via failure (i.e. error) avoidance is unlikely. Given the (unexpected) observation of an impulsivity-dependent response time effect, we repeated this analysis when including drug-induced response time changes. Here we observed that the modulation of SV by tyrosine and impulsivity was no longer significant (drug x level x impulsivity: $F_{(1, 19)}$ = 3.1,

p = 0.096). In sum, tyrosine enhanced the speed of difficult task performance in more impulsive participants, while also reducing their subjective value (or increasing the subjective cost) of difficult task performance. These findings suggest that the effects on SV do not reflect time-on-task avoidance, as faster instead of slower task performance was accompanied by lower SVs.

Finally, to assess whether the effect of tyrosine on cognitive effort discounting might reflect modulation of mood, we also assessed impulsivity-dependent effects of tyrosine on mood changes (total scores T1-T0), as assessed with the Bond and Lader analogue scale. Results of a repeated measures ANOVA showed no significant effects of tyrosine on mood changes (see Table 1), also not as a function of impulsivity (drug x impulsivity: F(1,25) = 2.7, p = 0.115).

### 3.4 Self-report N-back questionnaire and awareness of intervention

As expected, subjective ratings of perceived difficulty and effort increased as a function of N-back level, evidenced by a main effect of level on difficulty (F(3,81) = 136.3, p < 0.001) and effort (F(3,81) = 69.6, p < 0.001) rating. Also the perceived effort for completing higher N-back levels compared with level 1 increased linearly (F(2,54) = 71.1, p < 0.001). Consistent with our expectation, given that the task and these questionnaires were administered only shortly after the intervention, we do not observe any modulation of perceived difficulty or effort ratings as a function of drug (drug effect for difficulty: F(1,27) = 0.1; for effort: F(1,27) = 0.8; for relative effort: F(1,27) = 0.8), also not as a function of level (drug x level for difficulty: F(3,81) = 0.8; for effort: F(3,81) = 0.9; for relative effort: F(2,54) = 1.4). When entering trait impulsivity scores as a covariate, unlike the effect in the COGED paradigm, we do not observe any significant impulsivity-dependent drug effect on subjective effort ratings relative to the 1-back task, a measure most similar to the COGED task (drug x impulsivity: F(1,25) = 0.4, p = 0.539; drug x impulsivity x level: F(2,50) = 1.7, p = 0.198). This analysis strengthens the confidence that effects observed on the choice task are specific to the tyrosine intervention and were not observed on self-reported effort scores right after effort execution. After the second experimental session, participants reported their belief about the order of placebo and tyrosine sessions. Data is available for 23 out of 29 participants. 7 out of 23 participants judged the order correctly, 10 judged the order incorrectly and 6 refrained from giving any judgement because of a lack of confidence. Thus, the majority of these participants were unaware of the true intervention order.

## 4 Discussion

In this study, we set out to assess whether a catecholamine precursor alters motivation of cognitive control in older adults. More specifically, we hypothesized that augmenting catecholamine synthesis with tyrosine increases the subjective value of performing the N-back task for money. For this reason, we employed an established economic discounting procedure [16] that has previously been shown to be sensitive to cognitive load and aging. Our aim was to investigate tyrosine's effect on decision-making about cognitive effort, rather than tyrosine's role in N-back performance. Therefore, participants were exposed to the N-back task right after drug-intake (~20 min) at which point tyrosine should not have taken effect. Conversely, the effort discounting task was administered when catecholamine levels were expected to be enhanced.

In line with earlier reports, we observed that participants' performance decreased, and effort discounting increased, as a function of working memory load (i.e. N-back level)[16]. However, contrary to our prediction, tyrosine did not alter the subjective value of cognitive effort as a main effect. We predicted a positive main effect of tyrosine on motivation for

cognitive control, given prior evidence linking increased dopamine function with willingness to expend physical effort in animal models [5,84], with reduced physical effort discounting in humans [7] and neurocomputational models implicating striatal dopamine in increasing sensitivity to effort benefits versus costs [20]. Moreover, recent work has shown that increased dopamine might promote not just physical effort, but also cognitive control, by offsetting effort costs ([21,22]; but see e.g. [17–19] for domain-general and -specific effects of effort valuation). We expected that older adults might be particularly sensitive to benefits of tyrosine administration, given reports on reduced dopamine transmission [34] and, perhaps relatedly, reduced motivation to engage in effortful control [16]. Despite multiple lines of evidence that dopamine increases willingness to expend effort, our results indicate that tyrosine administration does not have simple uniform effects on motivation for cognitive effort across all participants.

Instead of a main effect, we observed in exploratory analyses an interaction with individual differences such that tyrosine effects depended on participants' baseline impulsivity. Specifically, the (demand-induced) subjective value of control decreased with tyrosine administration in participants with high baseline impulsivity, while it was increased, if anything, in less impulsive participants. This interaction between drug status and impulsivity is a small effect, but interesting because it mirrors a similar interaction between the catecholamine agonist methylphenidate and impulsivity in our previous study that had greater statistical power to assess individual differences (n = 100) [4]. In that study, participants with low impulsivity showed neutral or even reduced avoidance of cognitive demand, while those with high baseline impulsivity increased demand avoidance when given methylphenidate. The present results constitute an important extension of this prior work in two ways. First, they provide a critical conceptual replication of the result that catecholaminergic interventions can alter willingness to expend cognitive effort as a function of impulsivity despite differences in task, drug, and population. Second, we utilized a discounting task which explicitly measures cost-benefit decision-making, allowing us to directly test the hypothesis that pharmacological catecholamine manipulation modulates the subjective value of cognitive effort. The methylphenidate-dependent effects on demand avoidance in the prior study were plausibly linked to cost-benefit decision-making. However, the inference was indirect given that 1) no explicit rewards / benefits were on offer, 2) demand avoidance may have reflected ability to detect demand differences rather than increased sensitivity to effort costs, *per se* (cf. [13]) and 3) effort-execution (i.e. task-switching) in addition to choices were conducted on drug.

Our aim here was not only to extend previous work by teasing apart effort-execution and choices by administering them in two separate tasks, but also to show that tyrosine alters choices specifically without the possibility of performance modulation by having the N-back task performed off drug. Because of logistic reasons and plasma tyrosine levels reaching their peak level only after 90–120 minutes [27,40], we administered the N-back task shortly after tyrosine administration. Surprisingly, performance analyses indicate an impulsivity-dependent response time modulation which is also consistent with effects observed after methylphenidate administration [4]: Tyrosine attenuated level-related slowing in more impulsive participants. This finding evidences that, contrary to our expectation, tyrosine modulated performance as early as 20 minutes after tyrosine ingestion. Thus, as in the methylphenidate study, reduced motivation for cognitive effort (i.e. lower subjective value) on tyrosine was accompanied by level-dependent speeding, indicating, if anything, relatively better performance on the day of tyrosine administration. Although we remain puzzled by the fact that we observe any modulation of the effort execution phase by tyrosine that early after administration, the lack of changes in task performance (i.e. d') and the direction of response-time effects render a performance failure account unlikely. Self-report ratings of perceived effort

conducted right after effort execution support this interpretation, as ratings did not show (impulsivity-dependent) drug effects.

Moreover, the combination of response invigoration (perhaps indicating an increase in motor impulsivity) and lower subjective values (i.e. greater avoidance of monetary reward/ effort levels) would speak against the interpretation that subjective value changes were mediated by (effort-unrelated) changes in motor impulsivity/reward sensitivity.

In sum, the two studies converge in suggesting that dopamine interventions affect motivation for cognitive effort as a function of trait impulsivity, with undermining effects in more impulsive participants. Individual differences in trait impulsivity have been associated with baseline dopamine transmission [24] and drug effects on reversal learning [76], working memory [85], and striatal interconnectivity [86] in young adult samples. As such, instead of finding a main effect, our findings align with the proposal that the effects of catecholaminergic drugs vary with individual differences in baseline levels of dopamine [25].

## 4.1 Why does tyrosine reduce SV in more impulsive participants?

One reason that tyrosine might have attenuated the subjective value of cognitive effort is by paradoxically *decreasing* dopamine synthesis and dynamic dopamine response to offer presentation for more impulsive individuals, via D2 autoreceptors. Thus, a reduction in dopamine synthesis might result in a shift towards more cost and less benefit sensitivity. Indeed, while phasic DA release following offer presentation makes decision makers more sensitive to offer benefits and less sensitive to offer costs [20,87], autoreceptor binding can attenuate phasic DA release [88,89]. Thus, pharmacologically increased DA tone could, via increased autoreceptor signaling, reduce phasic DA release, making more impulsive decision-makers less willing to accept high-cost, high-benefit offers. This account assumes that tyrosine administration primarily increased pre-synaptic (autoreceptor) rather than post-synaptic D2 binding in older adults and that impulsive individuals differ in their pre-synaptic signaling sensitivity. Supporting the first assumption, the administration of phenylalanine, the precursor of tyrosine, to rats increased striatal dopamine release at lower doses, but attenuated dopamine release at higher doses [90]. In line with this finding, there is evidence for up-regulated striatal dopamine synthesis in older adults [91,92], which has been associated with impaired task-switching performance [92]. Thus, we speculate that a surge of precursor converted to dopamine in a system with already up-regulated dopamine synthesis may 'over-dose' the system triggering a shutdown of TH activity, via cytoplasmic dopamine or D2 receptors [93]. This would explain the observation that higher doses of tyrosine (both 150 and 200 mg/kg) were associated with reduced working memory performance compared with a lower dose [40]. In support of the second assumption, more impulsive individuals have lower presynaptic dopamine D2 receptor availability and greater amplitude phasic responses to instrumental cues at baseline [24]. Thus, more impulsive individuals may be particularly sensitive to the consequences of dopamine agonism for autoreceptor signaling. In sum, this account predicts that tyrosine reduced dopamine in older adults and more impulsive individuals will see the largest paradoxical reduction in phasic dopamine release with dopamine agonists. This mechanism has elsewhere been posited to explain why the agonist methylphenidate can reduce impulsive responding in individuals with ADHD [94].

Another reason that increased dopamine tone might increase cognitive effort discounting, for some people, relates to striatal dopamine's putative role in regulating action selection as a function of opportunity costs. In short, dopamine tone has been proposed to convey local environmental richness, and therefore the opportunity costs of 'sloth' [95,96]. Thus, in the competition between cognitive control, and habits, higher dopamine tone conveying higher

opportunity costs may promote an action selection bias for fast habits over slow control actions [97]. Indeed, strategic adjustments in the degree to which people perform fast and accurately on cognitive control tasks have been shown to depend on fluctuations in the average reward rate [98] and the strength of behavioral modulation by (background) average reward-rate was sensitive to pharmacological manipulations of the dopamine system [96,99]. This effect of increasing dopamine on action selection may also influence explicit cost-benefit decision-making about cognitive control of the type studied here [100]. Thus, individuals with high dopamine tone might be speculated to perceive their environment as particularly opportunity costly. In this context, tyrosine-related reductions in the value of cognitive effort as well as level-dependent speeding on the N-back task can be considered adaptive.

## 4.2 Limitations

Our hypotheses were motivated by a robust literature on dopamine's role in physical effort and cost-benefit decision making. However, tyrosine does not selectively increase dopamine: Oral administration in young adults has been shown to also affect plasma noradrenaline levels [101]. Future studies are needed to test the hypothesis that tyrosine alters motivation of cognitive control via affecting dopamine rather than noradrenaline transmission. This is especially pertinent because of the well-established link between the locus coeruleus–norepinephrine system and mental fatigue [102] and the implication of this system in task-related decision processes [103], task engagement and meta-cognitive regulatory functions [104,105]. Moreover, instead of a main effect of tyrosine administration on the subjective value of cognitive effort, we observe that tyrosine effects were modulated by trait impulsivity scores. Despite the fact that these results are consistent with our recent larger scale study (n = 100; [4]), we are aware that the effect is small and that our sample of 29 participants is low. As such, replication of this effect is advised in a larger sample, ideally in which the effort execution takes place before any intervention is administered. Lastly, the cognitive effort discounting paradigm we employed has several limitations. First, note that subjective values were extracted from participants' choices based on offers manipulating both, the monetary reward and effort level. Thus, intervention effects specific to effort cost and reward benefit sensitivity are not entirely dissociable in the current design. Second, we cannot fully exclude the alternative account that the effect of tyrosine on the subjective value of cognitive effort reflects an effect on risk aversion, given that the higher effort options were also associated with lower accuracy. Third, due to a profound delay of more than 2.5 hours between effort execution and choice phase, we cannot fully exclude the contribution of tyrosine effects on the memory of effort-level representations. Participants were exposed to the shape-effort associations multiple times during practice, a sheet of paper reminded them of the association during the choice task and we observed linear effects of effort-levels on subjective values, but the delay between effort execution and choices was larger here than in previous versions of this paradigm. Fourth, the employed paradigm does not allow participants to indicate preferences in favor of conducting a more effortful N-back level for an equal or lower monetary amount (i.e. cognitive effort seeking despite monetary "loss") and initial choices weight disproportionally more heavily towards the subjective value due to decreasing offer-amount adjustments. As such, replication of this effect using an experimental paradigm that takes into account the probabilistic nature of decision-makers and independently manipulates cost and benefits is recommended.

## 5 Conclusion

We demonstrate that tyrosine administration altered the subjective value of cognitive effort in healthy older volunteers (aged 60–75 years). However, contrary to our hypothesis that tyrosine

alters overall valuation of the N-back task, exploratory analyses suggest an interaction between drug and individual differences in trait impulsivity. Interestingly, as in our recent methylphenidate study, tyrosine reduced the motivation for cognitive effort in more relative to less impulsive participants. Thus, we show that cost-benefit decision-making about task engagement is sensitive to changes in catecholamine synthesis and the direction of effect depends on individual differences in trait impulsivity, a putative proxy of baseline dopamine function.

## Supporting information

**S1 File. Exclusion criteria.**
(DOCX)

**S2 File. Mood/Blood pressure T2-T0.**
(DOCX)

**S3 File. Questionnaire N-back.**
(DOCX)

**S4 File. Models.**
(DOCX)

**S5 File. Need for cognition and SV.**
(DOCX)

**S6 File. Control analyses.**
(DOCX)

**S7 File. Statistical effects of performance analyses.**
(DOCX)

**S8 File. Statistical effects of choice analyses.**
(DOCX)

**S9 File. Statistical effects of Model 1.2 now including 'offer amount' as random effect.** Note that this model gives convergence-warnings due to model complexity. In-text, we describe results of Model 1.2 presented in S8 File. However, this table shows that our main conclusions on impulsivity-dependent tyrosine effects hold.
(DOCX)

**S10 File. Choice trajectories.**
(DOCX)

## Acknowledgments

We thank Monique Timmer for medical assistance during data acquisition, Ilke van Loon and Ratigha Varatheeswaran for support in data collection and data processing, Payam Piray, Iris Duif, Joyce Dieleman in (tyrosine) sample preparation and Sabine Kooijman for advice on ethical approval procedure.

## Author Contributions

**Conceptualization:** Monja I. Froböse, Mirjam Bloemendaal, Esther Aarts, Roshan Cools.

**Data curation:** Monja I. Froböse, Mirjam Bloemendaal.

**Formal analysis:** Monja I. Froböse, Andrew Westbrook, Mirjam Bloemendaal.

**Funding acquisition:** Esther Aarts, Roshan Cools.

**Investigation:** Monja I. Froböse, Mirjam Bloemendaal.

**Methodology:** Monja I. Froböse, Andrew Westbrook, Mirjam Bloemendaal, Esther Aarts, Roshan Cools.

**Project administration:** Monja I. Froböse, Mirjam Bloemendaal.

**Resources:** Esther Aarts, Roshan Cools.

**Software:** Esther Aarts, Roshan Cools.

**Supervision:** Esther Aarts, Roshan Cools.

**Visualization:** Monja I. Froböse.

**Writing – original draft:** Monja I. Froböse, Andrew Westbrook, Roshan Cools.

**Writing – review & editing:** Monja I. Froböse, Andrew Westbrook, Mirjam Bloemendaal, Esther Aarts, Roshan Cools.

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
