## [Decision Letter · Decision Letter 0]

4 Sep 2019

PONE-D-19-19882

Catecholaminergic modulation of the cost of cognitive control in healthy older adults

PLOS ONE

Dear Mrs. Froböse,

Thank you for submitting your manuscript to PLOS ONE. After careful consideration, we feel that it has merit but does not fully meet PLOS ONE’s publication criteria as it currently stands. Therefore, we invite you to submit a revised version of the manuscript that addresses the points raised during the review process.

You will see that the reviewers found much to like about this study. However both reviewers raised issues about possible alternative interpretations of the observed patterns of results, which should all be addressed in a revision. Reviewer 1 notes that key findings regarding how opportunity costs relate to vigor (and cognitive effort exertion) should be discussed. This is important to address as opportunity costs are offered as a potential explanation for the observed changes in effort discounting.

We would appreciate receiving your revised manuscript by Oct 19 2019 11:59PM. To enhance the reproducibility of your results, we recommend that if applicable you deposit your laboratory protocols in protocols.io, where a protocol can be assigned its own identifier (DOI) such that it can be cited independently in the future. For instructions see: http://journals.plos.org/plosone/s/submission-guidelines#loc-laboratory-protocols

We look forward to receiving your revised manuscript.

Kind regards,

Ross Otto

Academic Editor

PLOS ONE

Journal Requirements:

Reviewers' comments:

Reviewer's Responses to Questions

**Comments to the Author**

1. Is the manuscript technically sound, and do the data support the conclusions?

Reviewer #1: Yes

Reviewer #2: Yes

2. Has the statistical analysis been performed appropriately and rigorously? 

Reviewer #1: Yes

Reviewer #2: Yes

3. Have the authors made all data underlying the findings in their manuscript fully available?

Reviewer #1: Yes

Reviewer #2: Yes

4. Is the manuscript presented in an intelligible fashion and written in standard English?

Reviewer #1: Yes

Reviewer #2: Yes

5. Review Comments to the Author

Reviewer #1: Froböse and colleagues present a Psychopharmacology study aimed at examining whether tyrosine modulates the willingness to exert cognitive effort in older adults. They use a well established previous task, in which effort is putatively manipulated with different levels of difficulty in an N-back task and participants are offered different magnitudes of monetary reward. They test older adults on this task in a double-blind, within-subject, placebo-matched crossover design and test whether the “SV” of exerting cognitive control (as measured using a decision-making task) is modulated by tyrosine. They find that Tyrosine does not modulate the willingness to exert effort per se, but in an exploratory analysis, show that there is an effect of the drug on the willingness to exert effort, but only as a function of trait impulsivity. They interpret their results as trait impulsivity being a marker of dopaminergic efficacy and tyrosine having different effects on motivation depending on levels of impulsivity.

This is a nice study, with appropriate measures, results that will be of interest to the field and atypically, I do not have many issues to raise with the analyses. I would also like to note that it is a nicely transparent study in terms of hypothesised vs exploratory results. My only real concern is about whether this study is actually finding results relating to “effort” or whether all of the results might be interpreted as a change in risk/impulsive preferences.

- The authors find that subjective value changes on Tyrosine vs placebo but only as a function of trait impulsivity. The interpretation is that this is a shift in the ‘cost’ of effort. However, given the nature of the cognitive control task, it is also plausible that this shift is actually more related to differences in risk preference, which may be at least as consistent with the current interpretation. Much of the literature they discuss suggests that D2/D3 in the dopamine system are linked to effort sensitivity, but the same system also regulates impulsivity and risk preferences (Buckholtz et al., 2010, Science). Moreover, there are several aspects of the data that suggest that effort may not be the cost that changes. First, the results show that impulsivity moderates the effect of Tyrosine, but, unless I missed it, ‘Need for cognition’ – which they label as an effort questionnaire – did not show a similar effect. Second, the N-back task has inevitably high error rates at higher levels of N and this perception of risk may have been what changes rather than the perception of effort. Thus, participants may have felt that there was a greater risk of error, or a greater risk of not getting the monetary reward, rather than cognitive control requiring more effort. Lastly, the confounding of effort and risk might be an alternative explanation of why other studies have shown an effect of dopaminergic manipulations on physical (Le heron et al., 2018; Brain) and cognitive effort (McGuigan et al., 2019, Brain) independently of trait measures, in tasks where the probability of getting rewarded is very high and constant across levels of effort. I therefore feel that the authors should discuss this alternative interpretation at length and also tone down interpretations of the effects being related to effort.

- In a slight contradiction to my previous point, the authors might want to highlight that there is some evidence in humans that the valuation of cognitive and physical effort occurs in not entirely overlapping systems in the brain (Chong et al., 2017, PLoS Bio). Moreover, in that study the same participants valued cognitive and physical effort differently, both mathematically, and also with only a very weak correlation between how motivated people were to exert effort one domain or the other. This would support the idea that cognitive effort may rely on distinct mechansims.

- The discussion of the effects being related to “opportunity costs” comes across as a little speculative. However, I think there is additional evidence that the authors could cite to support their claims, specifically Bierholm et al., (2013, Neuropsychopharmacology) and Le Heron et al., (2019, Bioarxiv).

Reviewer #2: This manuscript describes a study where the impact of tyrosine, a catecholamine precursor, was investigated on a paradigm that measures the subjective value of control. Specifically, a group of 29 elderly subjects (aged 60 to 75) underwent three sessions of which the last two contrasted the effects of tyrosine intake versus placebo on different levels of the n-back task as well as a subsequent valuation task. The authors did not find support for their main hypothesis, but observed an interesting interaction with Barratt Impulsiveness Scale-scores in exploratory follow-up analyses. The study is interesting, the method is sound, the analyses are well-introduced, and the results clearly discussed. I have no major concerns, but do have five questions / suggestions:

I have a question regarding the general paradigm, and wonder to which extent it might not be a limitation of this study. I was a bit surprised to see how much of an effect the first choice(s) has on the SV: “the magnitude of amount adjustments was cut in half after each adjustment”. I understand why this magnitude should reduce over the course of this task, but by making it 50% there is no option to correct a previous mistake. For example, if I was in doubt and accidentally gave the wrong response on the first trial, there is no possibility to reach my real SV over the remainder of this task? This makes the experiment quite sensitive to response errors/noise. Perhaps there are other reasons for this design feature that I am currently not seeing, but I was wondering if the authors could either briefly motivate this, or discuss this limitation (if they agree it is one). Maybe this is also something that could be checked or controlled for empirically (e.g., a participant that shows a pattern where all choices are opposite to their first could suggest something like this).

There is a bit of a delay between the N-back task and the effort discounting task (i.e., 3 hours: a break of 90 min, and an fmri session of 90 min). Was this also the case in previous versions of this paradigm, and is it not possible that some participants forgot the relation between some of the shapes and the different levels of difficulty? They are questioned about their experience, but only immediately after the n-back task. Perhaps, not their SV, but their memory of these different levels of difficulty is is further mediated by tyrosine and impulsivity?

The authors are correct in discussing the interaction with impulsivity scores with caution, given that it is not strong and comes from a set of secondary, exploratory analyses. However, this is currently not acknowledged in the abstract. For example, the authors could mention that: “Instead, in line with our previous study, *exploratory analyses showed that* drug effects varied...”.

It is mentioned that participants ate a very similar tasting banana yoghurt in the placebo condition. However, participants might not have tasted the difference, but still experienced a different sensation later during the session. Therefore, was awareness also tested, or does there exist other data on this from previous studies?

Figure 1 does not describe/depict the three paradigms (but the reference to this figure in the method section on page 10 suggests it does), or the 90 min break. Also, the lower timeline is a bit confusing as to whether it is depicting time (i.e., 20 min, 45 min, 189 min) relative to T0, or relative to the previous time point (i.e., 45 min is 45 + 20 min later than T0, etc.)?

Signed,

Senne Braem

6. PLOS authors have the option to publish the peer review history of their article (what does this mean?). If published, this will include your full peer review and any attached files.

Reviewer #1: No

Reviewer #2: Yes: Senne Braem

---

## [Author Response · Author response to Decision Letter 0]

16 Jan 2020

We would like to thank the reviewers for the time they spent reviewing our manuscript. They have offered constructive ways to further improve and clarify the manuscript. We took each of the concerns seriously and addressed them in two ways: First, we responded to each reviewer point and highlighted our actions following the reviewer comments (see the separate file: Response to Reviewers). Secondly, we made changes in the manuscript, supplemental material and figure 1 to correct the shortcomings that the reviewers perceived. These edits are pasted below the corresponding reviewer point (italic indent) and adapted in the according documents (see Revised Manuscript and Revised Supplements with tracked changes, changes underlined; Fig1_schematic_V2). Lastly, we provide an unmarked version of the revised manuscript and supplements (see Revised Manuscript and Revised Supplements).

We believe that the changes have considerably improved our paper, and we hope you agree.

---

## [Decision Letter · Decision Letter 1]

4 Feb 2020

Catecholaminergic modulation of the cost of cognitive control in healthy older adults

PONE-D-19-19882R1

Dear Dr. Froböse,

We are pleased to inform you that your manuscript has been judged scientifically suitable for publication and will be formally accepted for publication once it complies with all outstanding technical requirements.

With kind regards,

Ross Otto

Academic Editor

PLOS ONE

Additional Editor Comments (optional):

Reviewers' comments:

Reviewer's Responses to Questions

**Comments to the Author**

1. If the authors have adequately addressed your comments raised in a previous round of review and you feel that this manuscript is now acceptable for publication, you may indicate that here to bypass the “Comments to the Author” section, enter your conflict of interest statement in the “Confidential to Editor” section, and submit your "Accept" recommendation.

Reviewer #1: All comments have been addressed

Reviewer #2: All comments have been addressed

2. Is the manuscript technically sound, and do the data support the conclusions?

Reviewer #1: Yes

Reviewer #2: Yes

3. Has the statistical analysis been performed appropriately and rigorously? 

Reviewer #1: Yes

Reviewer #2: Yes

4. Have the authors made all data underlying the findings in their manuscript fully available?

Reviewer #1: Yes

Reviewer #2: Yes

5. Is the manuscript presented in an intelligible fashion and written in standard English?

Reviewer #1: Yes

Reviewer #2: Yes

6. Review Comments to the Author

Reviewer #1: I thank the authors for carefully addressing my points. I have no further concerns about this interesting piece of work.

Reviewer #2: Thank you for considering and responding to my suggestions/questions. I have no further suggestions, and like the additions that the authors made to the manuscript.

Signed,

Senne Braem

7. PLOS authors have the option to publish the peer review history of their article (what does this mean?). If published, this will include your full peer review and any attached files.

Reviewer #1: No

Reviewer #2: Yes: Senne Braem

---

## [Editor Report · Acceptance letter]

7 Feb 2020

PONE-D-19-19882R1 

Catecholaminergic modulation of the cost of cognitive control in healthy older adults 

Dear Dr. Froböse:

I am pleased to inform you that your manuscript has been deemed suitable for publication in PLOS ONE. Congratulations! Your manuscript is now with our production department. 

With kind regards,

on behalf of

Dr. Ross Otto 

Academic Editor

PLOS ONE